# Fast and quantitative 2D and 3D orientation mapping using Raman microscopy

Oleksii Ilchenko [1*], Yuriy Pilgun [2], Andrii Kutsyk[2], Florian Bachmann[3], Roman Slipets [1], Matteo Todeschini[4], Peter Ouma Okeyo[1,5], Henning Friis Poulsen [6] & Anja Boisen [1]

Non-destructive orientation mapping is an important characterization tool in materials science and geoscience for understanding and/or improving material properties based on their grain structure. Confocal Raman microscopy is a powerful non-destructive technique for chemical mapping of organic and inorganic materials. Here we demonstrate orientation mapping by means of Polarized Raman Microscopy (PRM). While the concept that PRM is sensitive to orientation changes is known, to our knowledge, an actual quantitative orientation mapping has never been presented before. Using a concept of ambiguity-free orientation determination analysis, we present fast and quantitative single-acquisition Raman-based orientation mapping by simultaneous registration of multiple Raman scattering spectra obtained at different polarizations. We demonstrate applications of this approach for two- and three-dimensional orientation mapping of a multigrain semiconductor, a pharmaceutical tablet formulation and a polycrystalline sapphire sample. This technique can potentially move traditional X-ray and electron diffraction type experiments into conventional optical laboratories.

[1] Department of Health Technology, Technical University of Denmark, Kgs. Lyngby, Denmark. [2] Faculty of Radio Physics, Electronics and Computer Systems, Taras Shevchenko National University of Kyiv, Kyiv, Ukraine. [3] Xnovo Technology ApS, Koge, Denmark. [4] Technical University of Denmark, National Center for Micro- and Nanofabrication, Kgs. Lyngby, Denmark. [5] Department of Pharmacy, University of Copenhagen, Copenhagen, Denmark. [6] Department of Physics, Technical University of Denmark, Kgs. Lyngby, Denmark. *email: olil@dtu.dk

Many materials are polycrystalline, meaning that they are composed of a large number of grains (crystallites) of different crystallographic orientations. A full three-dimensional (3D) orientation mapping of the grains (with information about the position, size, morphology, and orientation of each grain, as well as the topological connectivity between the grains) is crucial to relate structure to properties.

Electron backscattering diffraction (EBSD) is a widespread technique for such orientation mapping[1,2], but as a surface-based method, it can only be extended to 3D by serial sectioning. Moreover, it has challenges in the analysis of semiconductors and dielectric materials[1]. Transmission electron microscopy can provide 3D orientation maps with 1 nm resolution, but only in thin foils[3]. On the other hand, imaging based on X-ray diffraction using synchrotrons can provide 3D maps of thousands of grains in mm-sized samples with a resolution of 2 μm[4,5]—in scanning[6] or microscopy mode[7] even down to 100 nm—but this is an expensive and infrastructure-demanding technique. Recently developed mapping of grain orientations in 3D by laboratory X-rays (LabDCT) is a good alternative to the synchrotron measurements; however, its spatial resolution is limited to 10–15 μm[8].

Polarized Raman microscopy (PRM) has the potential to be developed into a relatively cheap orientation mapping technique[9], with a diffraction-limited spatial resolution of around 200 nm[10]. In PRM, if the symmetry of the crystal is known, Raman tensors $\Re_j$ for crystal modes $j$ can be obtained[11] and Raman intensities $I(\theta)$ versus sample rotation angle $\theta$ can be simulated[9]. Correlation between theoretical and experimental angular intensity dependencies $I_{theor}(\theta)$ and $I_{exp}(\theta)$ can, therefore, be used for the determination of the local crystallographic orientation[9] (see Supplementary Note 1). The first studies regarding polarized Raman spectroscopy began in 1964 when Loudon presented a Raman tensor analysis theory under the form of Raman scattering tensors for each of the 32 crystal classes[12]. Later, this theory was applied for crystallographic orientation analysis of inorganic[13–16] and organic single crystals[17,18], as well as inorganic crystals present in biological systems like calcite crystals in tergite exocuticles[19]. Raman tensor analysis theory was also developed and applied for the study of complex biomolecular crystals, including nucleobases in the nucleic acid of DNA[20,21], nucleosides[22,23], antihuman immunodeficiency virus agents[24], amino acids and peptides[17,25–27]. PRM has also been used for partially oriented biological systems like bone osseous tissues[28] and collagen fibrils in human osteonal lamellae[29]. During the last few years, the method has been applied to the study of crystallographic orientations, defects, doping effects, and Van der Waals interactions in 2D materials like graphene[30,31], MoTe$_2$[32], SnSe[33], ReS$_2$[34], MoS$_2$[35], GeAs[36,37], metal dehalogenases[35], and black phosphorous[35,38]. PRM has additionally been used for the investigation of GaP nanowires[39] and carbon multiwall nanotubes[40].

Most of the listed papers were focused on the determination of sample crystallographic orientations at the microscale, with the Raman microscope working in backscattering mode[9,41]. There has been a growing interest in confocal PRM studies of polycrystalline ceramics during the last years[42,43] including piezoelectric ceramics[44–46]. More recently an orientation type contrast was exploited in PRM studies of uranium oxide ceramics[47] and cancerous breast tissue[48]. In these papers, the PRM setup was used for improving the contrast in crystal features differentiation. The 3D domain orientation in BaTiO$_3$ crystals has been revealed after the sample has been mapped in a polarized Raman microscope with an a-plane crystal orientation[49]. Orientation distribution mapping of polycrystalline CuInSe$_2$ has also been demonstrated[10] by a combination of Raman contrast and EBSD mapping.

However, so far no one has presented a stand-alone orientation map based on Raman scattering, due to several experimental and theoretical problems. In particular, polarized Raman experiments have so far been carried out by rotating the sample or rotating a set of wave plates in the incident laser or on scattered beams in the Raman setup[9]. Such optical layouts require multiple Raman spectra measurements using numerous combinations of the orientation of the polarization state of the incident laser and orientation of the analyzer[41]. Due to this, Raman orientation mapping would need sequential polarization measurements at each raster-scanning point, which would make such hypothetical mapping procedures very complex and extremely slow and thus practically impossible. In addition, existing experimental PRM setups introduce ambiguities in measurements, implying that the approach fails completely for some orientations. As an example, in a scheme with normal sample illumination and on-axis collection of scattered light, it is impossible to determine a rotation of a (111) Si wafer around the [111] axis (Fig. 1a, Supplementary Fig. 1, Supplementary Note 1). Such problem can be partially avoided by tilting the sample or by realization of polarized off-axis Raman scattering registration[50] (Fig. 1b). However, as we show below, polarized Raman spectroscopy with off-axis registration also provides ambiguities in the determination of Euler angles, thus some combination of on-axis and off-axis channels must be used. To our knowledge, a theoretical investigation of the optimal number of on-axis/off-axis polarized Raman scattering channels in PRM has not been performed. Therefore, a measurement strategy for how to perform quantitative orientation mapping with an angular accuracy comparable with EBSD needs to be developed.

Here, using a concept of ambiguity-free orientation determination data analysis and simultaneous registration of multiple Raman scattering spectra obtained at different polarizations, our approach allows for 2D and 3D quantitative orientation mapping of multigrain materials. First results for silicon, a pharmaceutical tablet and for sapphire reveal favorable specifications: submicrometer resolution, fast data acquisition, and a high orientation resolution. The method applies to all Raman active materials independent of crystal symmetry and requires no sample preparation.

## Results

**Theoretical investigation of orientation ambiguity and error.** To assess the orientation determination ambiguity and error we have performed simulations on Si. Si has cubic symmetry (O$_h$ crystal class) and is well investigated by PRM[9,11] (Supplementary Note 2). We simulated a test dataset of Raman intensity variation versus wafer rotation angle $\psi$ for three types of Si wafers with surface oriented along (100) (Supplementary Fig. 2), (110) (Supplementary Fig. 3), and (111) (Supplementary Fig. 4) planes. The orientation determination results were obtained by fitting Euler angles to simulated data and were represented in a crystallographic color code using MTEX toolbox for Matlab[51,52], see Supplementary Fig. 5. The color corresponds to orientation of the wafer normal direction relative to crystallographic axes of Si. Misorientation angle, which corresponds to undetermined rotation of the sample, is depicted in Supplementary Fig. 5 as vertical bars on the right side of each subplot.

Orientation fitting was done using varying numbers of polarized channels (2–12 channels), see Supplementary Note 3, Supplementary Fig. 5. An analysis of the simulation results reveals, that basic orientation determination becomes possible for four or more channels. However, for some specific orientations, intensity data remains ambiguous and rotation angle $\psi$ is not recovered. This is clearly seen for the (111) wafer in misorientation plots in Supplementary Fig. 5, where the misorientation angle varies from zero to the largest possible for cubic symmetry

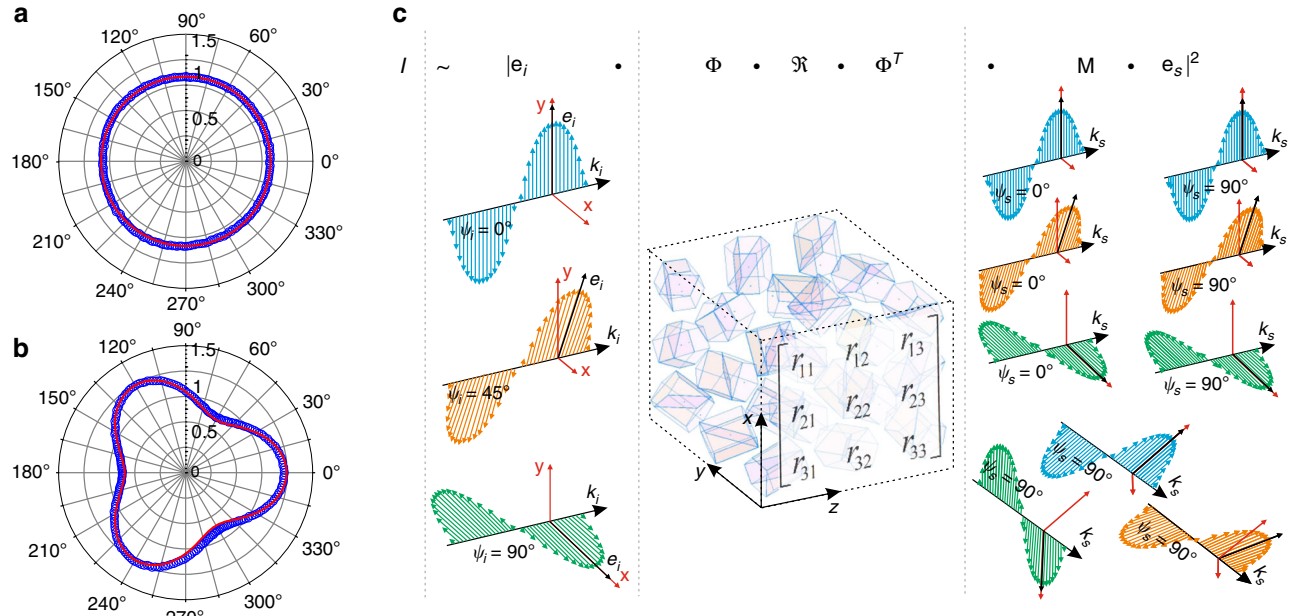

**Fig. 1 Optimization of the polarized Raman signal collection geometry.** Comparison between theoretical (red line) and experimental (blue circles) responses of the sum of Si modes versus wafer rotation angle $\psi$ plotted in polar coordinates for Si wafer with surface plane (111) for on-axis (**a**) and off-axis (**b**) polarized Raman scattering channels (**c**), illustration of the incident/scattering polarization and signal collection geometries for a solution with nine polarized channels.

**Table 1 Optimization of the orientation mapping configuration.** Optimization results for different numbers of channels and channel polarization combinations. Orientation error percentage represents geometric area on the sample, where reliable fit was achieved. Misorientation angle column represents ambiguity of the fit; the chosen number of channels was nine (highlighted in the red box).

| On-axis channels | | | | | | | | Off-axis channels | | | | Number of channels | Orientation error, % | Misorientation, ° |
|---|---|---|---|---|---|---|---|---|---|---|---|---|---|---|
| $\psi_i$ 0° | 45° | 90° | 135° | 0° | 45° | 90° | 135° | 0° | 45° | 90° | 135° | | | |
| $\psi_s$ 0° | | | | 90° | | | | 90° | | | | | | |
| ● | ● | ● | ● | ■ | ■ | ■ | ■ | ▲ | ▲ | ▲ | ▲ | 12 | 0.0 | <1 |
| ● | ● | ● | ● | ■ | ■ | ■ | ■ | ▲ | ▲ | ▲ | | 11 | 0.6 | <1 |
| ● | ● | ● | ● | ■ | ■ | ■ | ■ | ▲ | | ▲ | ▲ | 10 | 0.8 | <1 |
| ● | ● | ● | | ■ | ■ | ■ | ■ | ▲ | ▲ | ▲ | | 9 | 1.2 | <1 |
| ● | ● | ● | | ■ | ■ | ■ | | ▲ | | ▲ | | 8 | 4.7 | <1 |
| ● | ● | ● | ● | ■ | ■ | ■ | ■ | | | | | 8 | 48.1 | 62.8 |
| ● | ● | ● | ● | ■ | ■ | ■ | | | | | | 7 | 48.1 | 62.8 |
| ● | ● | ● | | ■ | ■ | ■ | | | | | | 6 | 48.0 | 62.8 |
| ● | ● | ● | | ■ | | ■ | | | | | | 5 | 55.7 | 62.8 |
| ● | ● | | | ■ | ■ | | | | | | | 4 | 63.5 | 62.8 |
| ● | ● | ● | | | | | | | | | | 3 | 75.2 | 62.8 |
| ● | | ● | | | | | | | | | | 2 | 88.2 | 62.8 |

62.8°. Nevertheless, even for such ambiguous data, some orientation data is still recovered, and it is possible to clearly distinguish wafers (100), (110), (111) even for four measurement channels. Full wafer orientation determination is only possible for nine or more channels.

Simulation results shown in Supplementary Note 4 and Supplementary Fig. 5 revealed that some orientations, like those appearing in the surface plane (111) of Si, are prone to introduce ambiguity in measurement data. Although simulation helped to define minimal number of polarization channels required to determine orientation, it still has limited scope, because only (100), (110), and (111) cases were considered.

To assess the quality of the orientation determination in the general case, additional analysis using exhaustive search of indistinguishable solutions over a range of all possible Euler angles was performed (described in details in Supplementary Note 4). That search revealed, that all measurement schemes with only on-axis channels applied to samples with cubic symmetry exhibit an ambiguity where two different orientations produce exactly the same intensity measurements. These two orientations have Euler angle $\psi$ which differs by 180°. Adding off-axis measurement channels removes this $\psi + 180°$ ambiguity.

Ambiguity simulation results for a multi-grain sample with different numbers of measurement channels are summarized in Table 1. The column "Orientation error" shows the percentage of the area where the local orientation is correctly determined. The maximum orientation error registered in the successfully fitted zones is shown in the column "Misorientation" in Table 1. The

same quantities versus total number of channels are plotted in Supplementary Fig. 6.

According to these simulations, the ambiguity can be successfully resolved when off-axis channels are added to the measurement setup. Still, adding only two off-axis channels is not sufficient to resolve ambiguity completely. Full resolution becomes possible when adding three off-axis channels to six on-axis channels, resulting in total nine measurement channels. Further addition of up to in total 12 channels does not noticeably increase the accuracy of the fit. Thus, in our experimental setup we decided to use nine channels.

The chosen geometry with nine polarized channels is illustrated in Fig. 1c. Three polarized laser beams $e_i$ with orientations of polarization state 0°, 45°, and 90° interact with a multigrain material with scattering properties dependent on local crystallographic orientation and described by the Raman tensor with rotation matrices $\Phi$ and $\Phi^T$. The scattered Raman signal $e_s$ (with corresponding rotation matrix $M$) from the three incident lasers is divided into nine backscattering channels (six on-axis and three off-axis) after propagation through three analyzers with orientations 0° and 90° for on-axis Raman scattering detection and 90° for off-axis Raman scattering detection. The information obtained from the nine Raman channels is used for determining the crystallographic orientation of a selected local volume. 2D or 3D orientation mapping is obtained by scanning the sample in $x$, $y$, $z$ with respect to the incident beams.

**Experimental setup**. Based on theoretical investigation of quantitative orientation mapping by Raman microscopy, we have developed and demonstrated a method, Single-Acquisition Raman Orientation Mapping (SAROM), for the study of crystalline samples. Our self-made Raman setup provides two orders of magnitude faster Raman polarization measurements compared with existing techniques and the ambiguity issue is overcome. The SAROM system is capable of simultaneously illuminating the sample with multiple laser beams at different orientations of the laser polarization state and detecting Raman scattering beams at multiple on-axis/off-axis scattering directions without using any moving parts (Fig. 2a, Supplementary Fig. 7). As shown in Table 1, chosen configuration with nine channels provides theoretically estimated accuracy of the orientation measurements by SAROM of <1°.

SAROM consists of a Laser Beam Delivery System (LBDS), which directs three laser beams with different orientations of polarization state on the sample focal plane (Fig. 2a), and an aberration-corrected Raman Beam Delivery System (RBDS) (Supplementary Figs. 8–12) with Wollaston Analyzer Unit (WAU), capable of splitting the Raman beam into on-axis and off-axis scattering geometries collected with different analyzer orientations (Fig. 2b–e, Supplementary Fig. 13). In a 2D mapping configuration, the sample is illuminated with a single laser source. Three laser beams with different orientations of polarization state are spatially separated on the sample and on the spectroscopic

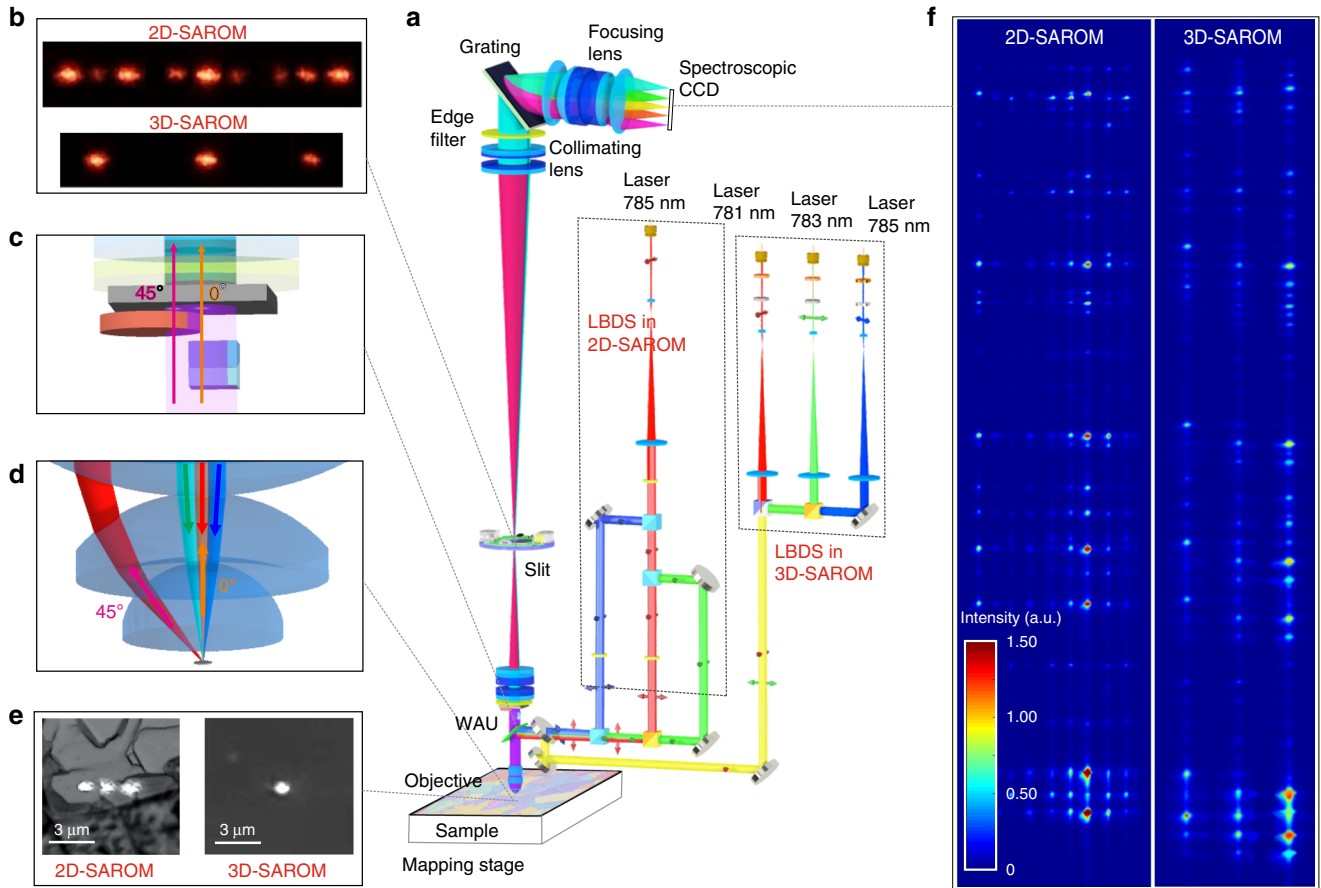

**Fig. 2 SAROM system design. a** Optical setup of SAROM in 2D and 3D mapping configurations designed in Zemax Optics Studio 17, **b** images of polarized beams on the spectroscopic slit focal plan, **c** on-axis and off-axis Raman scattering beam path through Wollaston Analyzer Unit (WAU), **d** sample illumination by laser beams and Raman signal collection geometry in on-axis (0°) and off-axis (45°) scattering pathways, **e** microscopy images of the laser spots on the sample in 2D- and 3D-SAROM configurations, **f** images of resulting Raman signals from carbamazepine drug obtained on a spectroscopic CCD in the 2D and 3D-SAROM configurations.

CCD focal planes (Fig. 2b-e). In a 3D mapping configuration, the sample is illuminated with three laser sources. Three laser beams with different orientations of polarization state overlap on the sample, however, they become separated on spectroscopic CCD focal plane (Fig. 1f) due to slightly different excitation wavelengths (see "Methods" section for details).

Artifact correction in polarized Raman measurements is a challenge. In order to reach an assessable error in SAROM, corrections are needed for: wavelength dependent intensity attenuation[53], Raman intensity scaling and normalization[43], Linear Phase (LP) and Linear Amplitude (LA) anisotropy of each optical element[54], and depth-dependent birefringence[13]. Since each channel has its own correction parameters, an experimentally based calibration becomes practically impossible. Therefore, we build a model, which predicts the values of the listed parameters (Supplementary Note 5, Supplementary Fig. 14).

Previously, the correction related to the Numerical Aperture (NA) of the microscope objective in PRM, has been realized by taking into account the full aperture of the microscope objective[11]. Here, we performed correction taking into account the NA values for on-axis and off-axis Raman scattering registration channels (see Supplementary Note 6, Supplementary Fig. 15).

**Application examples**. In the following, we demonstrate the method with a few examples. As an application in 2D, we present work on a polycrystalline Si (poly-Si) solar cell. The SAROM scanning procedure used for the poly-Si sample is shown in Supplementary Movie 1. We generated an orientation map of the poly-Si surface as shown in Fig. 3a by performing a least square fit to the Euler angles based on data from all nine polarized channels (Supplementary Note 7, Supplementary Figs. 16–18). In order to estimate the quality of the analysis technique, we compared this result with an orientation map of the same sample area using EBSD (Fig. 3b). A map of the local orientation difference: the misorientation angle (Fig. 3c) exhibited an average orientation difference of ~2.1° (Supplementary Figs. 19, 20). We argue that this error is dominated by a geometrical distortion of grains in the EBSD data (Supplementary Fig. 21). In order to minimize this effect we corrected the EBDS map for distortions and removed grain boundaries. Nevertheless, several artifact are still present on the map (Fig. 3c). Other sources of discrepancies may be connected with orientation dependent ambiguity of SAROM (Fig. 1, Supplementary Fig. 5) and an orientation determination error of EBSD.

Similar to EBSD[55], SAROM can be used for the study of extended defects like dislocations and grain boundaries, which e.g., directly correlate with the efficiency of solar cells.

Raman microscopy is often used for chemical mapping of pharmaceutical and biological materials[56]. It has been shown that PRM can be used to visualize particles on the surface of tablet formulations[57], however so far no quantitative information on the orientation of single particles has been provided. Here, we performed a content uniformity Raman measurement on a tablet containing carbamazepine dihydrate (CBZD) and polyvinylpyrrolidone (PVP). Using MCR, these components were decomposed including the fluorescence background as shown in Fig. 4a (Supplementary Fig. 22). Applying SAROM to the same area on the surface of the tablet, we obtained an orientation map of CBZD (monoclinic symmetry, $C_{2h}$ crystal class) (Fig. 4b, Supplementary Note 8, Supplementary Figs. 23–27, Supplementary Movie 2). These findings show the potential for SAROM to provide insight on crystal face functionality in pharmaceutical research and potentially in materials science[58].

Following the SAROM workflow (Supplementary Fig. 7a), we extended the method to 3D mapping of a semitransparent polycrystalline sapphire sample having grain sizes between 5 and 40 μm (Supplementary Note 9, Supplementary Fig. 28). 3D plots of the Raman spectra versus $\psi$ for different polarization configurations and correlations between theoretical and experimental responses of the $E_g(5)$ mode are shown in Supplementary Figs. 29–31, and in Supplementary Movie 3.

Nine maps of the intensity of $E_g(5)$ Raman mode, corresponding to the nine differently polarized Raman channels, are shown in Supplementary Fig. 35. Applying Supplementary Eq. 12 adapted to the trigonal symmetry of sapphire $D_{3d}^6$, we fitted the Euler angles at each measured point to these data (see Supplementary Note 10). The resulting 3D-SAROM volumetric orientation map is shown in Fig. 5 and in Supplementary Movie 4. 3D grain mapping is very important in ceramics technology, as properties such as fracture strength is strongly influenced by the statistical distribution of grain orientation and the grain boundary topology[59].

## Discussion

SAROM has some limitations. The theoretically determined orientation resolution varies with the orientation of the grain. Typically, the theoretical variation is well below 1° (Supplementary Fig. 6). Instrumental implementation implies that the error may increase up to 2°, which however still is sufficient for most applications. SAROM cannot be used on materials with high fluorescence background at the excitation wavelength of the laser source. However, the polycrystalline sapphire sample presented fluorescence at the grain boundaries, which we used as a perfect mark for grain visualization and exploited for further segmentation (Supplementary Figs. 32–34).

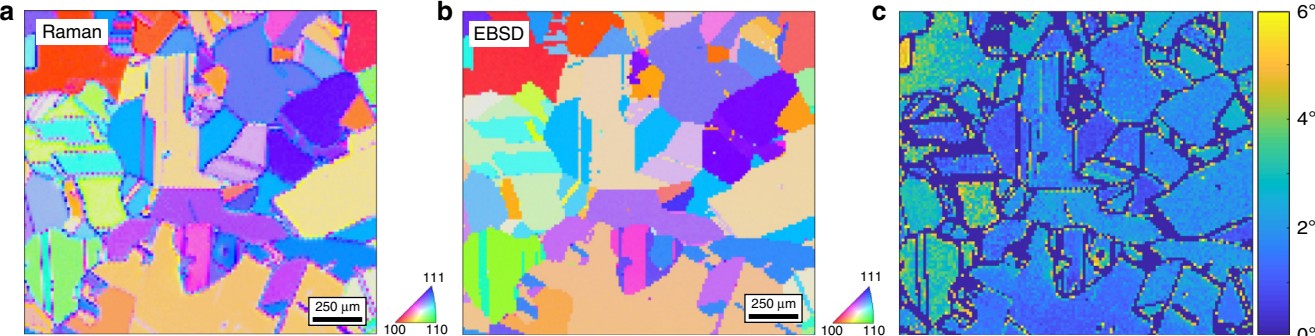

**Fig. 3 2D-SAROM results for Si. a** 2D-SAROM and **b** EBSD orientation maps of the surface of a polycrystalline Si sample. The colors refer to the orientations shown by the inverse pole figures (inserted). Both, SAROM and EBSD orientation maps have a step size of 13.51 μm, a map dimension of 131 × 120 pixels or 1770 × 1620 μm and an exposure time per step of 60 ms. **c** misorientation map between SAROM and EBSD data.

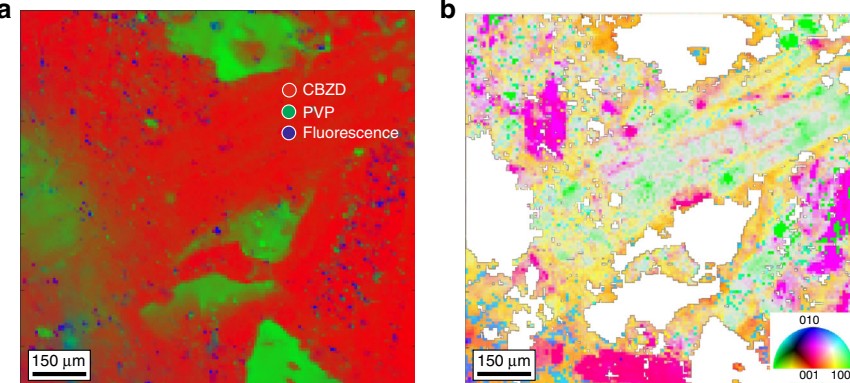

**Fig. 4 2D-SAROM for a compacted tablet. a** Chemical map obtained from the surface of a compacted tablet containing CBZD and PVP (2:1), **b** 2D-SAROM map showing the random orientation of CBZD particles on the surface of the tablet. The colors refer to the orientations as defined by the inverse pole figure (bottom right). SAROM was performed at a step size of 8 μm, map dimension 134 × 134 pixels or 1072 × 1072 μm, exposure time per step was 150 ms.

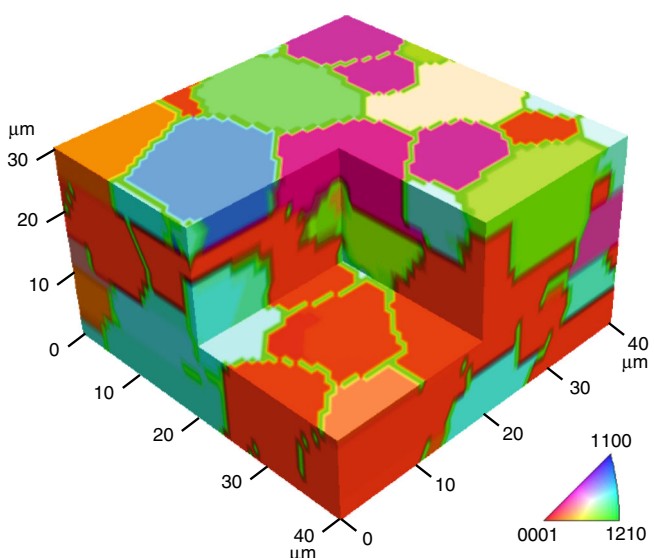

**Fig. 5 3D-SAROM on a polycrystalline sapphire sample.** The colors refer to orientations as shown by the inverse pole figure (inserted). The green boundaries between grains were generated as part of the segmentation procedure (Supplementary Figs. 32–34, Table S1). SAROM was performed at a lateral step size of 0.7 μm, axial step size of 1.7 μm. Map dimension is 57 × 56 × 18 pixels or 40 × 39.2 × 30.6 μm with an average exposure time per step of 180 ms.

The 3D-SAROM analysis has several limitations. The depth of scanning is limited by the material transparency and Raman scattering cross-section[45] of the sample at laser/Raman scattering wavelengths: for the polycrystalline sapphire study the maximum depth is ~0.1 mm at 785 nm laser wavelength. In-depth SAROM mapping is also limited by off-axis laser refraction effects leading to Raman signal attenuation and decreased axial resolution[60–63]. In order to minimize these effects we used oil immersion microscope objective for 3D measurements of polycrystalline sapphire (Supplementary Fig. 36). These limitations of in-depth scanning can be partly overcome by increasing the spectra acquisition time.

Further development of SAROM may include the use of visible range lasers. In comparison with near infrared lasers, used in the present study, visible lasers will increase the Raman cross-section and the SAROM spatial resolution. However, the choice of excitation sources should also take into consideration the transparency of material at the excitation wavelength and possible resonance Raman effects which may change the Raman tensor coefficients leading to errors in orientation determination or modifications in Raman tensor analysis[64].

In summary, SAROM is a nondestructive quantitative orientation mapping method with a high diffraction-limited spatial resolution, similar to that of confocal Raman microscopy (Supplementary Fig. 37). It applies to all Raman active inorganic and organic crystalline material. Due to the simultaneous measurement of multiple polarized channels, we have demonstrated fast scanning capabilities: 16 min for the 2D orientation maps of polycrystalline Si (131 × 120 pixels), 46 min for the 2D orientation map of tablet formulation (134 × 134 pixels) and 176 min for the 3D orientation map of polycrystalline sapphire (57 × 56 × 18 pixels). Furthermore, since it is a confocal method, it does not require complex tomographic data reconstruction as in the case of 3D X-ray orientation microscopy[65]. The equipment has a relatively low price compared with 3D-EBSD and high-resolution 3D X-ray microscopy[66]. SAROM is applicable to a broad range of problems in 2D materials, polymers, drugs, and biomolecular research. It is useful in mineralogy, geology, semiconductors (solar cells, microelectronic substrates), ceramics (piezo-, magneto-, and ferroelectrics), and superhard materials (abrasives, drilling tools, superhard transparent windows). Moreover, the functionality of SAROM can be extended by combination with e.g., confocal Raman microscopy, hyperspectral imaging, or polarized light microscopy. Further development of SAROM can be expanded into instantaneous polarized Raman mapping similar to wide-field Raman microscopy[67,68].

The developed method is therefore likely to become a simple, economically accessible, and broadly used characterization tool for 2D/3D crystallographic mapping. Moreover, it expands the range of materials that can be analyzed and could bring new insight into our understanding of the structure of matter.

## Methods

**Materials**. Silicon wafers with surface planes (100), (110), (111) were provided by National Center of Micro- and Nanofabrication in Denmark.

Polycrystalline Si solar cell was provided by the factory Pillar (Kyiv, Ukraine).

Anhydrous carbamazepine was obtained from Sigma-Aldrich (CAS No.298-46-4). Povidone (polyvinylpyrrolidone, PVP, K60) was obtained from Sigma-Aldrich (CAS No. 9003-39-8).

Sapphire monocrystalline plates at a-, c-, m-, and r-planes were purchased from Crystran Ltd.

Polycrystalline sapphire sample with dimensions 10 × 5 × 5 mm and average grain size of 20 μm was purchased from CoorsTek GmbH.

**EBSD analysis of poly-Si**. The surface of the polycrystalline Si was mechanically polished prior to EBSD investigation. Silica microparticles having dimensions of 6, 3, and 1 μm were sprayed on a rotating diamond grinding disc for the polishing.

Starting with the larger microparticle size, the sample was alternately polished and visually inspected using an optical microscope. The inspection was necessary to verify that the surface was without scratches along a particular direction, index of an inhomogeneous polishing. The same polishing procedure was repeated using each microparticle size.

The polished sample was mounted on an EBSD holder with a 54° tilt respect to the electron beam perpendicular. The holder was installed on a FEI Nova 600 NanoSEM stage and the sample was tilted 16°, to reach a total tilt of 70° respect to the electron beam perpendicular. The microscope was equipped with a Bruker QUANTAX EBSD detector and operated at an accelerating voltage of 15 kV, a beam current of 10 nA, a 40 μm aperture and at a working distance of 12 mm. The detector was positioned in such a way that the smallest distance between the electron-beam focusing point at the specimen surface and the camera was 16.5 mm. All measurements were performed in high vacuum mode. EBSD orientation map was collected with a pattern resolution of $320 \times 240$ pixels, exposure time of 155 ms, and step size of 25 μm, total measurement time 3.3 h. The raw data were processed using MTEX toolbox in Matlab.

**Recrystallization of carbamazepine dihydrate, tablet preparation, and solid-state analysis.** Anhydrous carbamazepine was dissolved (25 mg/mL) in a vial filled with a water–ethanol solvent composition (3:1) at 60 °C. The vial was then removed from the hot plate and allowed to cool down to room temperature. After around 4 days, needle and plate shaped particles of CBZD formed via slow evaporation crystallization and were harvested by filtration.

CBZD was identified via PANalytical Xpert PRO X-Ray Diffractometer (PANanalytical B.V., Almelo, Netherlands) with a nickel filtered CuKα ($\lambda = 1.5418$ Å) source that was generated at a tube voltage of 45 kV and current of 40 mA. A PIXcel detector with a $\theta/\theta$ goniometer was used. The measurements were performed at a reflection mode between 5° and 35° at a step size of 0.0263° $2\theta$ and a scan speed of 0.06734° $2\theta$. The X'Pert Data Collector software provided by PANalytical, Almelo, Netherlands was used in order to analyze the diffractograms. For reproducibility purposes, triplicate measurements were performed.

In order to visualize the molecular structure (SF2), intermolecular interactions, and different faces (SF3) of CBZD, Mercury CSD 3.10.3 (Build 205818) was used. Mercury was provided by the Cambridge Crystallographic Data Centre, UK.

A tablet press (Gamlen Tableting Ltd, UK) with a 6 mm diameter cylindrical flat-faced compacts was used in order to prepare the tablets. This tablet press was equipped with a 500 kg load cell 8CT6–500–022). 200 mg (±10 mg) of CBZD and 100 mg (±10 mg) were compacted at a compression speed of 60 mm/min. The tableting die was lubricated with magnesium stearate suspension prepared in acetone. Tablets were prepared in triplicate.

X-ray powder diffraction (XRPD) was used in order to confirm carbamazepine (CBZ) to be carbamazepine dihydrate (CBZD). In order to achieve this, the experimentally obtained XRPD diffractogram of CBZD was compared with the calculated diffractogram (CBZD, FEFNOT02, monoclinic, P21/c)[69] from the Cambridge Structural Database (CSD).

**Working principle of SAROM.** In order to obtain high-quality 2D/3D Raman orientation maps, we developed and experimentally verified all required data acquisition and data analysis steps. The overview of SAROM workflow is shown in Supplementary Fig. 7. It consists of six different steps: (i) the development and construction of the Raman setup, which is capable to acquire simultaneously nine polarized Raman spectra from each scanning point on the sample; (ii) the development of preprocessing and artifacts correction algorithms applied to the polarized Raman spectra; (iii) the development and verification of the Raman tensor model; (iv) single acquisition Raman orientation mapping at multiple polarization channels; (v) fitting of Euler angles at each mapping point and (vi) the color code-based representation of the orientation map.

**Simultaneous measurement of polarized Raman spectra.** One of the most critical steps which needed to be solved in SAROM is achievement of high speed of polarization measurements. For this purpose, we designed and constructed a Raman setup without moving parts, capable to illuminate the sample with multiple laser beams at different laser polarizations and simultaneously detect Raman scattering beams at multiple analyzer orientations (Supplementary Fig. 8). The working principle of the simultaneous measurement with nine polarized Raman channels is schematically shown in Supplementary Fig. 7b. Before starting the optical design having such complicated structure, we performed test measurements in a 'traditional' polarized Raman microscope layout[41] in order to find the minimum number of polarization measurements required for obtaining valid crystal orientations (Table 1). The 'traditional' optical setup can acquire one polarized Raman spectrum per measurement[9]. Therefore, it requires a set of sequential measurements with different combinations of incident laser polarization angle $\psi_i$ and analyzer angle $\psi_s$. In order to get more flexibility in the definition of incident laser/analyzer orientation we slightly modified the common used Porto notations[54] for the back scattering on-axis and off-axis measurement configuration to the following form: $z(\psi_i, \psi_s)\bar{z}$.

We performed test simulations and data fitting on the surface of monocrystalline Si wafers with surface planes (100), (110), and (111), respectively.

It was determined that at least three differently polarized laser orientations and three analyzer orientations are required, with mandatory presence of off-axis channels. In total, we got nine combinations of polarization measurements which can be defined as $z(\psi_i^1 \psi_s^1)\bar{z}$, $z(\psi_i^1 \psi_s^2)\bar{z}$, $z(\psi_i^1 \psi_s^{\text{off-axis}})\bar{z}$, $z(\psi_i^2 \psi_s^1)\bar{z}$, $z(\psi_i^2 \psi_s^2)\bar{z}$, $z(\psi_i^2 \psi_s^{\text{off-axis}})\bar{z}$, $z(\psi_i^3 \psi_s^1)\bar{z}$, $z(\psi_i^3 \psi_s^2)\bar{z}$, $z(\psi_i^3 \psi_s^{\text{off-axis}})\bar{z}$. The angles $\psi_i$, $\psi_s$ generally can have different values, however in the case of Si analysis with $O_h$ symmetry we used the following angles: $\psi_i^1 = 0°$, $\varphi_i^2 = 45°$, $\psi_i^3 = 90°$, $\psi_s^1 = 0°$, $\psi_s^2 = 90°$, $\varphi_s^{\text{off-axis}} = 90°$. We chose these values as a result of optimization (see Supplementary Note 4, Table 1). Simultaneously obtained experimental data in SAROM setup are shown in Supplementary Figs. 2–4. In general, our system can work at any $\psi_i$, $\psi_s$ which can be beneficial for a specific crystal symmetry. If required, SAROM is capable to work with more than nine simultaneously acquired polarized channels. The actual number of channels is limited only by the optical design of the system and spectroscopic sensor dimensions.

**Laser beam delivery system (LBDS).** The technical realization of the three differently polarized laser spots on the sample focal plane can be performed with two different laser beam delivery optical layouts: 2D- and 3D-SAROM configurations. LBDS in 3D-SAROM consists of three thermally stabilized diode lasers $L_1$, $L_2$, $L_3$ operated at slightly different wavelengths 781 nm, 783.5 nm and 784.8 nm, respectively. Wavelength of each laser can be finely adjusted by the change of diode temperature in order to get the best spectral distance between vibrational modes. The laser beams have polarization angles $\psi_i^1 = 0°$, $\psi_i^2 = 45°$, $\psi_i^3 = 90°$ with respect to the $x$ axis in laboratory coordinate system $(x, y, z)$ (Supplementary Fig. 8). The beams are collimated and centered in the optical layout and then focused at the same spatial point on the sample focal plane. Raman scattering responses from the three differently polarized laser beams are then divided on the spectroscopic CCD focal plane due to the spectral difference in the laser frequency, which is around 20 $cm^{-1}$ (Fig. 2f). 3D-SAROM is beneficial for 3D mapping and for materials with low number of phonon modes, where Raman peaks overlap does not lead to significant data analysis problems.

In 2D-SAROM, three differently polarized laser channels were formed via splitting of the beam from one laser source with a wavelength of 785 nm. Three laser beams then focused at different coordinates on the sample focal plane (Supplementary Fig. 8). Similar concept was demonstrated in polarization-resolved Raman measurements in liquids[70].

Comparing with 3D-SAROM, spectral profiles in 2D-SAROM are not overlapped at spectroscopic CCD. Therefore, 2D-SAROM is beneficial for 2D mapping with complex Raman spectra, where peaks overlapping can lead to uncertainties in vibration mode responses.

On the other hand, 2D-SAROM needs separate rows on the spectroscopic CCD for the registration of Raman signals from differently polarized laser beams. In the case of 3D orientation measurements in 2D-SAROM configuration, the diffraction limited laser points become blurred and overlapped, due to reflection index-caused aberrations when the laser is focused in the depth of the material[49]. Therefore, for the 3D orientation mapping of polycrystalline sapphire we decided to use 3D-SAROM configuration. In such a way, the responses from differently polarized laser spots become non-overlapped on spectroscopic CCD.

**Raman beam delivery system (RBDS).** Another big challenge in SAROM setup is connected with the design of the Raman beam delivery optical path, including Wollaston Analyzer Unit (WAU), which provides simultaneous measurements at three different analyzer orientations: 0° and 90° in on-axis registration and 90° in off-axis registration.

The key module of RBDS is the WAU. It consists of quartz Wollaston prism, analyzer, and mask (Supplementary Fig. 13). Wollaston prism splits on-axis Raman scattering into two analyzer configurations 0° and 90°, off-axis Raman beam pass through the analyzer. Mask works as a spatial filter for improved separation between on-axis and off-axis channels.

After the WAU, Raman beams are coupled by slit focusing lens with a self-designed and patent pending imaging spectrograph (application number PCT/DK2019/050027) (Supplementary Fig. 9). It is a lens-based spectrograph with a transmission fused silica grating. This grating provides almost polarization-independent spectral efficiency at the level of 96%, which leads to the minimized artifacts in polarized Raman measurements and high sensitivity of the spectrograph[53]. Unique feature of the spectrograph design consists in combination of low NA collimation lens with high NA focusing lens. These lenses provide magnification equal to 0.2×, which leads to the compression of the Raman beam energy down to the size when an entire Raman spectrum covers only one row on the spectroscopic CCD (Supplementary Fig. 36). The spectrograph focusing lens is a self-designed achromatic double-gauss type lens, which consists of six spherical elements (Supplementary Fig. 10). It has a diffraction limited spot size through the entire focal plane, equal to the size of the spectroscopic CCD (30 mm), which provides excellent imaging conditions required for the multichannel SAROM setup. The overall photograph of the SAROM setup is shown in Supplementary Fig. 11.

**Linear phase and linear amplitude effects in SAROM setup.** The most critical problem in optical path is related to the polarization artifacts compensation for

each polarization channel. For this purpose, we performed numerical polarization analysis using Zemax Optics Studio and obtained estimation about LP and LA anisotropy parameters for propagation through each optical element (Supplementary Fig. 14). In order to minimize LP effects produced by the dielectric mirrors and dichroic beamsplitters during beam propagation from the laser source to the detector and from the sample focal plane to the WAU, we optimized the geometry of beam propagation as shown in Supplementary Fig. 14. In this layout, the phase shift between the S and P components of the Raman scattering vector $\mathbf{e}_s$ produced on the first reflective surface was compensated by the next reflective surface and finally delivered to the sample with minimized artifacts.

LA effects are mostly produced by the dichroic mirrors DM1, DM2 and which leads to the rotation of $\mathbf{e}_s$ in the range of $0-2.2°$, depending on the original $\mathbf{e}_s$ orientation. This deviation was compensated by corresponding choice of polarizations during least square fitting of Euler angles.

**Custom microscope objective**. In order to provide efficient simultaneous off-axis and on-axis Raman scattering signal registration we designed and constructed custom Raman microscope objective (Supplementary Fig. 12). Zoomed area in Supplementary Fig. 12A illustrates the difference between collecting cones of light scattered in the Silicon sample at on-axis and off-axis geometries. Immersion-type objective allows performing off-axis measurements at 45° with regard to the surface normal. However, due to high reflective index of Si, real collected scattering angle from depth of the sample was around 15°. We confirmed resulting internal angle by the fit of off-axis experimental data on Si wafer with plane (111) (see Supplementary Fig. 4E).

**Optical path and components description in SAROM**. Here we provide a detailed description of the Raman system, where all components are listed at its appearance in the optical path from the source to the detector. The names of the component correspond to those in Supplementary Fig. 8.

As an excitation sources a single mode diode lasers from Thorlabs LD785-SE400 (785 nm, 400 mW) were used.

The laser clean-up filters Lf1-Lf4 (Semrock, cat. no. LD01–785/10–12.5) are placed on the beam path after the laser output in order to block unwanted emission background from the laser pumping, reducing its intensity of about six orders of magnitude.

In 2D-SAROM setup, the collimated beam from laser was expanded in beam expander (two NIR coated spherical lenses, f1 = 20 mm (Edmund Optics, cat. no. 45–792) and f2 = 100 mm (Edmund Optics, cat. no. 45–806)) up to the diameter of 10 mm. Expanded beam was divided into three beams by a set of beamsplitters BS1-BS3 and polarized BS1 (Edmund Optics, cat. no. 49–005, Edmund Optics, cat. no. 49–870).

In 3D-SAROM setup, we used three diode lasers stabilized at different temperatures in order to control laser wavelength. Each laser beam propagates through separate beam expanders (lenses f1, f2). Final beam diameter for three laser beams was 10 mm. All beams combined by polarized beamsplitter BS2 (Edmund Optics, cat. no. 49–870) and non-polarized beamsplitter BS4 (Edmund Optics, cat. no. 49–005).

The collimated beams from LBDS in 2D and 3D configurations reflect from mirrors M1-M6 (Thorlabs, cat. no. BB1-E03) and pass through a motorized power attenuation filter wheel (Standa, cat. no. 10MWA168) with seven optical density (OD) filters inside.

SAROM can be commutated to the 2D-SAROM or 3D-SAROM geometries by switching of the motorized flipper mirror MFM (Newport Corp., cat. no. 8893-K-M, mirror from Edmund Optics, cat. no. 63–145).

Then the laser beams reflect from dichroic mirror DM1 (Semrock, cat. no. Di02-R785–25x36) and passes through the dichroic mirror DM2 responsible for the coupling with a visible microscope. Finally, laser beams are delivered to the custom made microscope objective (NA = 1.2, focal length 4.1 mm, Supplementary Fig. 12).

The microscope is equipped with a white light LED illumination unit (Thorlabs, cat. no. MNWHL4) and imaging CCD (ToupTek, cat. no. E3CMOS02300KPB). White light from the LED collimates and passes through the edge filter (Semrock, cat. no. FF01–650/SP-25), in order to block the NIR part of the LED emission spectrum. In such a way a visible real-time image can be monitored during the registration of Raman spectra (see Supplementary Movie 1). After that, a white-light beam is reflected by 2-in. mirror M7 (Thorlabs, cat. no. BB2-E03) and focused at the back focus of the microscope objective using a focusing lens f7 (Edmund Optics, cat. no. 47–317). Commutation between illumination and light reflection from the sample is organized via beamsplitter BS5 (Thorlabs, cat. no. BSW10R). The sample illumination unit of the microscope is combined with Raman channel via short-pass dichroic beamsplitter DM2 with cut-off wavelength 749 nm (Semrock, cat. no. FF749-SDi01–25 × 36 × 3.0).

The collected Raman beam passes through microscope objective and DM1, DM2 to the slit lens f3 (Edmund Optics, cat. no. 47–271). Lens f3 is a NIR achromatic doublet pair which focuses Raman beam on the motorized spectrograph slit (Newport, cat. no. 77738).

Spectrograph consists of NIR achromatic doublet pair lenses f4 (Thorlabs, cat. no. AC508-750-B), custom designed Double-Gauss focusing lens f5, transmission fused silica polarization independent grating (Light Smith) and edge filter Ef2

(Semrock, cat. no. BLP01-785R-50). Raman spectra collected on the air cooled CCD (Andor, iDus 416).

Aberration corrected design of the spectrograph and double-gauss lens design are shown in Supplementary Figs. 9 and 10. Spectrograph design is covered by the patent of Technical University of Denmark (application number PCT/DK2019/050027). Unique protected feature of the spectrograph design consists in combination of low NA collimation lens with high NA focusing lens. Focusing lens is a double-gauss type lens which consists of six spherical elements. It is corrected on coma, astigmatism and spherical aberration in the entire focal plane equal to the size of CCD (30 mm). The quality of objective can be verified by MTF curves (Supplementary Fig. 10E, F). Point spread function adjusted to the size of CCD pixel (Supplementary Fig. 10C). Spectrograph stray light parameters were optimized using nonsequential analysis mode in Zemax Optics Studio 17. It was experimentally verified that stray light level at wavenumber shift of 30 cm$^{-1}$ from laser line was at the level of $2.1 \times 10^{-4}$ which is very good result taking into account relatively small spectrograph dimensions and focal lengths. Spectrograph inside view is shown in Supplementary Fig. 9B.

**SAROM measurement conditions**. In the case of SAROM microscopy of mono- and polycrystalline Si laser power on the sample in each polarized channel was 50 mW, laser spot size was 3 μm, exposure time per mapping point was 0.05 s, CCD readout mode was multitrack.

In the case of SAROM microscopy of CBZD drug laser power on the sample in each polarized channel was 10 mW, laser spot size was 3 μm, exposure time per mapping point was 0.15 s, CCD readout mode was multitrack.

In the case of SAROM microscopy of mono- and polycrystalline sapphire laser power on the sample in each polarized channel was 100 mW, laser spot size was 0.8 μm, exposure time per mapping point was varied from 50 ms to 2 s (exponentially scaled depending on the mapping depth), CCD readout mode was multitrack.

## Data availability
The source data for Figs. 3–5 are provided with the paper. The data that support the other findings of this study are available from the corresponding author upon reasonable request.

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

## Acknowledgements
We would like to acknowledge Chaoling Xu for help with acquiring EBSD data and Erik Lauridsen for discussions related to the analysis of 3D Raman data and color code based visualization of the orientation maps. We would like to acknowledge Yury Gogotsy and Asia Sarycheva for conceptual advices regarding manuscript structure. Center for Intelligent Drug Delivery and Sensing Using Microcontainers and Nanomechanics (IDUN) funded by the Danish National Research Foundation (grant no. DNRF122) and the Velux Foundations (grant no. 9301), Proof of Concept grant (33009) of the Technical University of Denmark.

## Author contributions
O.I. invented SAROM principles, designed and built the SAROM microscope, analyzed the data, performed the experiments and wrote the paper, Y.P. analyzed and resolved ambiguities in SAROM, developed algorithms for Raman tensor analysis and color code data visualization, A.K. developed algorithms for Raman tensor analysis and color code data visualization, F.B. identified SAROM ambiguities and supported misorientation analysis, R.S. wrote the software for the Raman microscope and supported in the experiments, M.T. performed EBSD measurements of poly Si, P.O.O. supported pharmaceutical tablet related experiments, H.F.P. and A.B. gave conceptual advice and revised the manuscript

## Competing interests
The authors declare no competing interests.
