## [Article File - Peer Review File · Nature Communications]

Reviewers' comments:

Reviewer #1 (Remarks to the Author):

The manuscript "Fast and quantitative 2D/3D orientation mapping using Raman microscopy" by Ilchenko et al. is a nice report about the application of Raman microspectroscopy for the mapping of local crystal orientations. The authors invested a lot of efforts not only in describing the experimental setup and conduction of the measurements; they also provided datasets and videos giving very nice insight into, e.g., how the acquisition of the Raman hyperspectral images works. I would recommend the publication of this work. However, there are some issues to be addressed in a revised version of the manuscript:

1) The authors give various examples using scales rather on the order of 100 μm (according to the scale bars). It is not clear what the pixel or step sizes are for each Raman map shown; would be good to add this information to each Raman map. Now one question from the readership may be whether the presented approach exhibits any limitation in terms of spatial resolution. What experience have the authors made regarding Raman orientation mapping with resolutions on the submicrometer scale?

2) Related to the previous item: the authors use rather high laser powers of 10-100 mW. The authors of Ref. 10 of the present manuscript reported about damages by laser powers of larger than 4 mW (in the case of CuInSe₂ thin films). Does the proposed methods exhibit any limitation in terms of laser power?

3) When comparing the Raman and EBSD maps in Figure 2, there are discrepancies visible in terms of the false colors (and with that of the local orientations) for some grains. It would be good to add a line of explanation to the text about where these differences in false colors come from.

4) A minor issue, page 6, line 206, "Silicon wafers with orientations (100), (110), (111)": "(hkl)" does not indicate orientations but surfaces; the authors should rephrase this sentence: "silicon wafers with surface planes of (100), (110), (111)". The authors should correct also similar expressions throughout the main text / supplementary materials.

Reviewer #2 (Remarks to the Author):

The manuscript reports on an original Raman set-up allowing the recording of maps showing the distribution of the orientation of crystals. The concept is not original from the science point of view

but the set-up is new and offers a large potential for material science. The manuscript is well organized and deserves publication after three weaknesses have been addressed.

- Previous uses of Raman “2D” mapping to get information on crystalline orientation is much larger than reported in the manuscript and more references must be given (see some below: 1-7).

- Obtaining information from inside of the matter, the so-called 3D mapping, is not obvious and the limitations must be addressed and appropriate references given (see some below: 8-11)

- The problems of multiphased materials, in particular those made of poor and good Raman scatterers must be also addressed (see refs below).

Both the optical phenomena inside the matter (modification of the optical index distorts the volume of focused light) and the huge difference between the Raman effect of different phases made of more or less covalent bonds (e.g. a very poor scatterer like alpha alumina and very good scatterers like ZrO₂ or GdAl₂O₃, ref 12) make 3D mapping may be efficient only for some phases.

1. Bone osteonal tissues by Raman spectral mapping: Orientation–composition, M. Kazanci, P. Roschger, E.P. Paschalis, K. Klaushofer, P. Fratzl, *Journal of Structural Biology* xxx (2006) xxx–xxx

2. <https://www.witec.de/resources-and-education/knowledge-base/show/advanced-raman-techniques-and-combinations/how-should-one-perform-polarization-dependent-measurements>: Polarization dependent measurements allow the investigation of polarization dependent activity of specific peaks in a Raman spectrum. This can be used to analyze molecular orientations and geometries of e.g. crystal lattices, liquid crystals, amorphous materials, or polymers.

3. 3D Raman mapping of the collagen fibril orientation in human osteonal lamellae, Susanne Schrof, Peter Varga, Leonardo Galvis, Kay Raum, Admir Masic, *Journal of Structural Biology*, 187(3), 2014, 266-275.

https://www.researchgate.net/publication/236925231_Polarized_Raman_Anisotropic_Response_of_Collagen_in_Tendon_Towards_3D_Orientation_Mapping_of_Collagen_in_Tissues

4. Calcite distribution and orientation in the tergite exocuticle of the isopods *Porcellio scaber* and *Armadillidium vulgare* (Oniscidea, Crustacea) – a combined FE-SEM, polarized SCM-RSI and EBSD study, Bastian H. M. Seidl, Christian Reisecker, Sabine Hild, Erika Griesshaber, Andreas Ziegler. *Z. Kristallogr.* 2012, 227, 777–792 /DOI10.1524/zkri.2012.1567
<https://www.degruyter.com/downloadpdf/j/zkri.2012.227.issue-11/zkri.2012.1567/zkri.2012.1567.pdf>

5. Polarized Raman backscattering selection rules for (hhl)-oriented diamond- and zincblende-type crystals J.A. Steele , Pascal Puech , R.A. Lewis , *Journal of Applied Physics* 120, 055701 (2016);
<https://doi.org/10.1063/1.4959824>

6. Drug <http://www.nicoletcz.cz/upload/kc/files/raman/drxxi/Raman%20Polarization.pdf>

7. M. Pham Thi, G. March, P. Colomban, Phase Diagram and Raman Imaging of Grain Growth Mechanism in Highly Textured $\text{Pb}(\text{Mg}_{1/3}\text{Nb}_{1/3})\text{O}_3\text{-PbTiO}_3$ Piezoelectric Ceramics, *J. Eur. Ceram. Soc.* 25 [4] (2005) 3335-3346.

8. G. Gouadec, L. Bellot-Gurlet, D. Baron, P. Colomban. Raman mapping for the investigation of nanophased materials. Zoubir A. Raman imaging. Techniques and Applications, 168, Springer, pp.85-118, 2012, Springer Series in Optical Sciences, 978-3-642-28251-5. 10.1007/978-3-642-28252-2_3. https://link.springer.com/content/pdf/10.1007%2F978-3-642-28252-2_3.pdf

9. Modeling and measuring the effect of refraction on the depth resolution of confocal Raman microscopy , Everall, NJ *APPLIED SPECTROSCOPY* 54 (6) 773-782, 2000

10. In-depth analyses by confocal Raman microspectrometry: experimental features and modeling of the refraction effects , Bruneel, JL; Lassegues, JC; Sourisseau, C , *JOURNAL OF RAMAN SPECTROSCOPY* 33(10), 815-828 Published: OCT 2002

11. Confocal Raman microspectrometry: A vectorial electromagnetic treatment of the light focused and collected through a planar interface and its application to the study of a thin coating, Sourisseau, C; Maraval, P , *APPLIED SPECTROSCOPY* 57(11) 1324-1332 Published: NOV 2003

12. Ruby micro-piezospectroscopy in $\text{GdAlO}_3/\text{Al}_2\text{O}_3(\text{/ZrO}_2)$, $\text{Er}_3\text{Al}_5\text{O}_{12}/\text{Al}_2\text{O}_3(\text{/ZrO}_2)$ and $\text{Y}_3\text{Al}_5\text{O}_{12}/\text{Al}_2\text{O}_3(\text{/ZrO}_2)$ binary and ternary directionally solidified eutectics , Gouadec, Gwenael; Makaoui, Karim; Perriere, Loic; et al., *JOURNAL OF THE EUROPEAN CERAMIC SOCIETY* 32(10) 2145-2151 Published: AUG 2012

Reviewer #3 (Remarks to the Author):

The manuscript reports the development of an advanced device for polarization-resolved Raman microspectroscopy. A 2D and a 3D scheme are presented for the purpose of mapping the orientation of crystallites in polycrystalline samples. Proof-of-concept experiments were carried out using a Si wafer, a Si-based solar cell, a tablet, and a sapphire material.

Overall, the presented device is novel and very interesting. It has the potential to add to the analysis of polycrystalline materials and partly replace existing (in particular X-ray-based) methods. Moreover, it means an important advancement of Raman microspectroscopy instrumentation. However, the principles behind the concept have been known for quite some time and they have been used for obtaining very similar (if not the same) pieces of information, see e.g. ref 10. In other words, the manuscript's novelty is more at the instrument end than in terms of methodology. Therefore, its suitability for a general science journal is limited and I would recommend submitting a

revised version to a more specialized journal, e.g. Journal of Raman Spectroscopy, Applied Spectroscopy, or Review of Scientific Instruments.

That said, the manuscript has also a number of shortcomings that need to be addressed before it can be considered for publication.

1) The description of the method and the setup needs to be improved and made easier to understand as only part of the readership will be familiar with Raman scattering in detail. In its current form, the text is somewhat confusing, e.g. as it is occasionally not clear which angle is actually meant: polarization state or scattering angle.

2) The concept utilized in the 2D scheme is not particularly new. Irradiating neighboring spots with beams of different polarization and then imaging the emission to a spectrograph was already demonstrated for polarization-resolved Raman measurements in liquids, see e.g. Analytical Chemistry 2017, 89, 11, 5725-5728.

3) The uncertainty owing to the hole diameters in the masks of the Wollaston analyzer unit is not addressed/assessed. The diameter is directly related with the angular range, over which the signals are collected and hence it should have an impact on the results.

4) The current method is called "fast" but it still requires scanning the sample stepwise. Do you see the potential to do an instantaneous mapping like in standard light microscopy? This may be worth commenting on in the conclusion section.

Response to reviewers comments regarding “Fast and quantitative 2D/3D orientation mapping using Raman microscopy” manuscript

Answers to reviewer comments:

Reviewer #1:

1) The authors give various examples using scales rather on the order of 100 μm (according to the scale bars). It is not clear what the pixel or step sizes are for each Raman map shown; would be good to add this information to each Raman map. Now one question from the readership may be whether the presented approach exhibits any limitation in terms of spatial resolution. What experience have the authors made regarding Raman orientation mapping with resolutions on the submicrometer scale?

Answer. We agree with the importance of adding information about the details of mapping. In the revised version of the manuscript, we have added step size, mapping dimensions and exposure time under the description of each figure.

Our minimum step size in the lateral dimension was 0.7 μm when we measured the sample of polycrystalline sapphire. This was very close to the diffraction limited spot size for laser excitation of 0.785 μm at objective NA=0.95 in the medium of sapphire ($n=1.76$ at 0.785 μm).

It is still possible to decrease the step size which will improve the resolution, but this will lead to linearly increased total measuring time. In order to reach a compromise between acceptable resolution for sapphire grains (average size was around 25 μm) and total measuring time we used lateral step size of 0.7 μm and axial step size of 1.5 μm (which was also close to the axial diffraction limit).

However, if SAROM system in the future is constructed with 2-3 times improved diffraction limited spatial resolution by the usage of blue or green lasers with high NA microscope objectives (for example, NA=0.95 for dry objectives and NA=1.4 for immersion objectives) then it will be possible to carry out orientation mapping at resolution of 200nm. This will be our next step in the development of SAROM. This point has been added in the discussion section in the revised version of the manuscript.

In conclusion, we do not see any differences in the estimation of SAROM spatial resolution in comparison with limitations in confocal Raman microscopy.

2) Related to the previous item: the authors use rather high laser powers of 10-100 mW. The authors of Ref. 10 of the present manuscript reported about damages by laser powers of larger than 4 mW (in the case of CuInSe₂ thin films). Does the proposed methods exhibit any limitation in terms of laser power?

Answer. In our experiments, we were trying to use laser at high power in order to reach the balance between acceptable signal-to-noise ratio in Raman spectra and fast mapping capabilities. The chosen samples; silicone, carbamazepine and sapphire were not damaged by the laser due to relatively low absorptivity at laser excitation of 785nm. However, laser power can be decreased to the level of few mW (or 100 μ W) if necessary. Of course, this will lead to increased exposure times if the experiment requires orientation determination accuracy at the level of few degrees or better.

3) When comparing the Raman and EBSD maps in Figure 2, there are discrepancies visible in terms of the false colors (and with that of the local orientations) for some grains. It would be good to add a line of explanation to the text about where these differences in false colors come from.

Answer. The discrepancies in false color has different sources of origin. First of all, they relate to the mismatch of the geometrical shape between SAROM and EBSD grains. SAROM map was conducted under the normal incidence of laser with regard to the sample surface and signal was collected in backscattering schema. Therefore, SAROM map has no geometrical distortions. However, EBSD map was collected at 54° tilt with the electron beam being perpendicular to the sample. Therefore, silicon grains were distorted. This geometric misfit between maps is clearly seen in supplementary Fig. S20. We were trying to correct distortions in EBSD maps in Matlab, however, some misfit was still present. In order to avoid this effect we removed the boundaries on the final misorientation map. Nevertheless, several artifacts were still present.

A second source of discrepancy is the mathematical ambiguity of orientation determination. Supplementary Fig. S10 shows that ambiguity is different for different surface planes of Si. Since, actual map of polycrystalline Si consists of crystals with different planes, SAROM reveals different accuracy in orientation determination (usually less than 1 degree).

A third source of discrepancy can be the error of orientation determination in EBSD map. According to the specifications of the EBSD system, we expect that it was at the level of 1 degree (average over all Euler angles).

We agree with the reviewer regarding the necessity to add mentioned aspects of discrepancies into the main text. Please see additional text marked by yellow color in the revised version of the manuscript.

4) A minor issue, page 6, line 206, "Silicon wafers with orientations (100), (110), (111)": "(hkl)" does not indicate orientations but surfaces; the authors should rephrase this sentence: "silicon wafers with surface planes of (100), (110), (111)". The authors should correct also similar expressions throughout the main text / supplementary materials.

Answer. We appreciate the reviewer for pointing out this issue. It was corrected through all the main and supplementary text.

Reviewer #2:

1) *Previous uses of Raman “2D” mapping to get information on crystalline orientation is much larger than reported in the manuscript and more references must be given (see some below: 1-7).*

Answer. We agree with the reviewer and we thank for providing us with more references. We have included additional references connected with lateral mapping by polarized Raman microscopy in the revised version of the manuscript. It includes references 1-7 provided by the reviewer as well as several other references where applications on 2D materials, nanowires and nanotubes, ferroelectrics and biological crystals were demonstrated.

2) *Obtaining information from inside of the matter, the so-called 3D mapping, is not obvious and the limitations must be addressed and appropriate references given (see some below: 8-11)*

Answer. We agree with the reviewer and we have added text in the discussion section and text in supplementary material (**Section S10. Poly-sapphire 3D mapping**) with several references (including references 8-11 provided by reviewer).

Comment. It was a pleasure for us to read three papers (ref. 9-11 provided by reviewer) because they actually support our consideration about the choice of correct microscope objective for 3D mapping. Motivated by the reviewer comments we performed a simulation of lateral (Fig. S35f, S35g) and axial (Fig. S35d, S35e, S35h, S35i) resolution for typically used high resolution dry metallurgic objective in confocal Raman microscopy (Fig. S35a) and self-designed oil immersion objective for 3D measurements in Raman microscopy (Fig. S35b). For the correct comparison, as a dry objective, we have chosen an infinity-corrected, semi-apochromatic microscope objective having an N.A. of 0.85 at laser beam diameter 6.5mm which provides a magnification of 50 times when used with a telescope objective having an effective focal length of 183 mm (Olympus patent US04417787) (Fig. S35a). Our self-designed objective has similar parameters: NA=0.85 at laser beam diameter of 8.8mm, magnification 45x at telescope objective having an effective focal length of 183 mm (Fig. S35b).

Therefore, both objectives provided equal lateral resolution on the sample surface (black lines in Fig. S35f, S35g). However, Huygens point spread function (PSF) simulation has shown that axial resolution becomes significantly worse in the case of dry objective in the depth of sapphire (-50 μ m) (Fig. S35h) rather than for an oil immersion objective (Fig. S35i). Taking into account the values on the scale bar in Fig. 35d-35i, it is possible to conclude that aberration driven signal attenuation in the depth of 50 μ m is around 55% for dry metallurgic objective and 11% for oil immersion objective.

During 3D mapping of sapphire using oil immersion objective (Fig. S35b) we observed much more signal attenuation than 11% due to signal absorbance and reflections on grains in polycrystalline sapphire. As we have stated in supplementary material (**Section S10. Poly-sapphire 3D mapping**) the exposure time on the surface of sapphire was 50ms and we were increasing it exponentially up to the level of 2s for the last depth layer at $-30\mu\text{m}$ in order to reach constant spectrum quality for each depth layer (constant signal-to-noise ratio). Therefore, the total Raman signal attenuation from the surface to the depth of $30\mu\text{m}$ was 40 times. This was a really challenging sample for Raman measurements (see Fig. S27b with the photograph of the turbid structure of polycrystalline sapphire sample).

We agree with the reviewer that real situation of PSF in a polycrystalline sample is not obvious. Simulation performed in Fig. S35 only takes refraction, diffraction, and spherical aberration effects into account. However, in the future, we should add more parameters like sample transparency and reflection on the grain boundaries. On the other hand, we believe, that we have chosen an optimal microscope objective (Fig. S35b) for 3D measurements for the polycrystalline sapphire sample. Generally, limitations of 3D mapping by SAROM are the same as for confocal Raman microscopy. Therefore, an in-depth analysis of this problem would be better to present in a stand-alone manuscript.

Fig. S35. Point spread function (PSF) of microscope objectives. a, an infinity-corrected, semi-apochromatic dry microscope objective (N.A.=0.85 at laser beam diameter 6.5mm,

magnification 50x, **b**, self-designed oil immersion objective for 3D measurements in Raman microscopy (NA=0.85 at laser beam diameter 8.8mm, magnification 45x), **c**, illustration of air-sapphire interface with regions of simulated PSF, **d**, axial PSF of dry objective in Fig. S35a in air, **e**, axial PSF of oil immersion objective in Fig. S35b in oil, **f**, cross sections of lateral PSF of dry objective (Fig. S35a) in air (black line) and in the depth of sapphire at -50 μ m (red line), **g**, cross sections of lateral PSF of oil immersion objective (in Fig. S35b) in oil (black line) and in the depth of sapphire at -50 μ m (red line), **h**, axial PSF of dry objective in Fig. S35a in the depth of sapphire at -50 μ m, **i**, axial PSF of oil immersion objective in Fig. S35b in the depth of sapphire at -50 μ m.

3) *The problems of multiphased materials, in particular those made of poor and good Raman scatterers must be also addressed (see refs below).*

Answer. We agree with the reviewer that multiphased materials will create extra problems for SAROM. Especially, if the sample requires 3D mapping: each material will create extra losses of Raman signal due to reflections and increased aberrations. A discussion of this issue and the limitation of SAROM related to poor Raman cross sections of multiphase materials have been added in the discussion section in the revised version of the manuscript.

4) *Both the optical phenomena inside the matter (modification of the optical index distorts the volume of focused light) and the huge difference between the Raman effect of different phases made of more or less covalent bonds (e.g. a very poor scatterer like alpha alumina and very good scatterers like ZrO₂ or GdAl₂O₃, ref 12) make 3D mapping may be efficient only for some phases.*

Answer. We agree with reviewer that non-destructive 3D-SAROM measurements will be limited by refraction, diffraction, spherical aberrations and sample transparency. These limitations have been added in the discussion section in the revised version of the manuscript.

As a recommendation, with regard to reviewer comment, we suggest to increase spectrum collection time if Raman cross section of the investigated sample is relatively low. In such a way, SAROM would not be a “fast” technique but it would still be able to provide quantitative orientation mapping at limited depth. Also, it is possible to perform SAROM, with blue or green lasers which will increase the Raman cross section. This will be an interesting solution if the sample does not absorb much light and does not produce much fluorescence under the excitation wavelength, which interfere with the Raman spectrum. It is also important to mention that in polarized Raman microscopy we should avoid resonance Raman effect, usually induced under the excitation by blue or green lasers which may lead to a modification of the Raman tensor of the material. This could create additional source of error in orientation determination.

Reviewer #3:

However, the principles behind the concept have been known for quite some time and they have been used for obtaining very similar (if not the same) pieces of information, see e.g. ref 10.

Answer. We believe that the most important achievement in our work is the fact that we demonstrate quantitative orientation determination by Raman microscopy. Previously obtained Raman experiments **i)** have never combined normal and tilted detection of scattering light for further Raman tensor analysis, **ii)** corresponding ambiguity analysis of orientation determination has never been performed and **iii) ambiguity free** solution has never been proposed. For example, in ref. 10 (Schmid, T., et al. Orientation-distribution mapping of polycrystalline materials by Raman microspectroscopy. *Sci. Rep.* 5, 1–7 (2015)) authors were using only two polarized channels for orientation mapping by Raman microscopy. Based on our investigation, the orientation determination error for such method was at the level of 25-50% as shown in Fig. 1 in the revised version of the manuscript. Therefore, authors in ref 10 were trying to fit polarized Raman contract data from the maps acquired at orthogonally polarized laser states with EBSD data in order to obtain orientation. However, independent orientation mapping by Raman microscopy with accuracy comparable with EBSD (around 1 degree in the full range of Euler angles) has never been demonstrated before.

Our theoretical work to assess the ambiguity and error of SAROM was presented in the supplementary material in the original manuscript. In order to highlight our results in **quantitative** orientation mapping we have now added a section called “Theoretical investigation of orientation ambiguity and error” in the main part of the manuscript (results section).

1) *The description of the method and the setup needs to be improved and made easier to understand as only part of the readership will be familiar with Raman scattering in detail. In its current form, the text is somewhat confusing, e.g. as it is occasionally not clear which angle is actually meant: polarization state or scattering angle.*

Answer. We agree with the reviewer that method and setup description was not clear for the broader audience. Revised version of the manuscript includes a broader introduction to the method and SAROM setup which hopefully will make it easier to understand.

In the updated version we separated polarization angle and scattering angle definitions in a way that polarization state angle is called “angle of polarization state” and scattering angle is called “off-axis scattering registration angle”.

2) *The concept utilized in the 2D scheme is not particularly new. Irradiating neighboring spots with beams of different polarization and then imaging the emission to a spectrograph was already demonstrated for polarization-resolved Raman measurements in liquids, see e.g. Analytical Chemistry 2017, 89, 11, 5725-5728.*

Answer. We agree with the reviewer that irradiating neighboring spots with beams of different polarization and further imaging of Raman signal to a spectrograph was already demonstrated in *Analytical Chemistry* 2017, 89, 11, 5725-5728. Therefore, we have included this reference in the

updated version of the manuscript (in the Methods section). Nevertheless, we believe that many aspects of the SAROM principle are novel, in particular the fit between the theoretical search of the geometrical configuration of polarized channels for minimized orientation determination error and the experimental realization of this search in a working prototype with an orientation determination accuracy at the level of 1 degree.

3) *The uncertainty owing to the hole diameters in the masks of the Wollaston analyzer unit is not addressed/assessed. The diameter is directly related with the angular range, over which the signals are collected and hence it should have an impact on the results.*

Answer. We thank the reviewer for pointing out this issue. In order to make a valid analysis of the impact of hole diameters to the uncertainty of orientation determination we performed a simulation.

We use holes (apertures) for on-axis and off-axis Raman scattering collection that have a certain NA value. SAROM maps of poly-Si, CBZD and poly-sapphire were collected within a cone with aperture half-angle of 24°. This effect becomes critical especially for high NA objectives¹ relevant for confocal 3D-SAROM measurements. Taking into consideration this angular distribution, Raman intensity can be described in the following way:

$$I_{\Omega} \propto \sum_j \int_{\Omega_i} \int_{\Omega_s} \left| \mathbf{e}_s'^T \mathbf{J}_s^T \mathbf{\Phi}^T(\theta, \phi, \psi) \mathfrak{R}_j(X, Y, Z) \mathbf{\Phi}(\theta, \phi, \psi) \mathbf{J}_i \mathbf{e}_i \right|^2 d\Omega_i d\Omega_s, \quad (\text{S10})$$

where Ω_i, Ω_s – solid angles of incident and scattering irradiation, respectively. Integration is done within the range of cones formed by optical setup for incident and scattered path.

Effect of collecting scattered light from different angles due to integration over aperture can be estimated analytically for uniform intensity response of the collecting lenses. Registered intensity is proportional to integral

$$I_{\Omega_s} \propto \int_0^{2\pi} \int_0^{\theta_s} \left| \mathbf{e}_s'^T(\theta, \varphi) \mathfrak{R} \mathbf{e}_i' \right|^2 \sin \theta d\theta d\varphi, \quad (\text{S11})$$

where θ_s is maximal polar angle of collection cone. It is convenient to calculate this integral in local spherical coordinate system (Fig. S37), where z axis is oriented along center collection direction and polarization \mathbf{e}_s' lays in the XOZ plane.

Fig. S37. Local spherical coordinate system associated with collection aperture.

Scattered polarization \mathbf{e}'_s is perpendicular to wave vector \mathbf{k}_s and explicitly depends on integration angles θ and φ . Components of vector \mathbf{e}'_s can be found with the help of rotation matrices

$$\mathbf{e}'_s = \begin{pmatrix} \cos \varphi & -\sin \varphi & 0 \\ \sin \varphi & \cos \varphi & 0 \\ 0 & 0 & 1 \end{pmatrix} \begin{pmatrix} \cos \theta & 0 & \sin \theta \\ 0 & 1 & 0 \\ -\sin \theta & 0 & \cos \theta \end{pmatrix} \begin{pmatrix} \cos \varphi & \sin \varphi & 0 \\ -\sin \varphi & \cos \varphi & 0 \\ 0 & 0 & 1 \end{pmatrix} \begin{pmatrix} 1 \\ 0 \\ 0 \end{pmatrix}. \quad (\text{S12})$$

Raman tensor \mathfrak{K}' rotated to local coordinate system and incident polarization vector \mathbf{e}'_i remain constant during integration. One can define result of their tensor multiplication as a vector

$$\mathbf{p}_R = \begin{pmatrix} p_x \\ p_y \\ p_z \end{pmatrix} = \mathfrak{K}' \mathbf{e}'_i. \quad (\text{S13})$$

To make comparison of aperture-integrated intensity we normalize intensity to integral over solid angle

$$\Omega_s = \int_0^{2\pi} \int_0^{\theta_s} \sin \theta d\theta d\varphi = 2\pi(1 - \cos \theta_s) \quad (\text{S14})$$

Normalized intensity integral (S11) transforms to

$$I_s \square \frac{1}{2\pi(1 - \cos \theta_s)} \int_0^{2\pi} \int_0^{\theta_s} |\mathbf{e}'_s{}^T \mathbf{p}_R|^2 \sin \theta d\theta d\varphi. \quad (\text{S15})$$

For specific angle θ_s this integral can be computed analytically, resulting in

$$I_s \square \frac{1}{48} (33 + 12 \cos \theta_s + 3 \cos 2\theta_s) p_x^2 + \frac{1}{48} (3 - 4 \cos \theta_s + \cos 2\theta_s) p_y^2 + \frac{1}{48} (12 - 8 \cos \theta_s - 4 \cos 2\theta_s) p_z^2 \quad (\text{S16})$$

For small θ_s most intensity comes from term p_x^2 , which is in accordance to used selection of local coordinate system. For increasing θ_s coefficient near p_x^2 decreases, and contribution from p_z^2 correspondingly increases. Effect of p_y^2 is much smaller and can be neglected even for as large θ_s angles as 30° .

For transparent samples the collection angle value should be corrected for refractive index of a material, when signal collection is done from internal sample volume. Silicon has quite large index of refraction $n=3.8$ at 785 nm, so internal collection angle decreases significantly comparing to external collection angle. For an aperture which collects light from a cone with half-angle of 24° , internal collection angle will be limited to $\theta_s = 6^\circ$. Corresponding contributions from different components of vector \mathbf{p}_R are

$$I_s \propto \sum_j (0.9971 p_x^2 + 1.37 \times 10^{-6} p_y^2 + 0.0029 p_z^2) \quad (\text{S17})$$

Our estimations show, that integration over the holes which produce a collecting cone with aperture half-angle of 24° will lead to a small change of registered intensity due to intermixing of tensor product components. Resulting change will be smaller than 0.3% for the silicon sample, so this effect can be safely ignored in that case. For CBZD and poly-sapphire samples intensity variation caused by component intermixing will be of order 1.5% due to lower refractive index. It can be corrected if required.

A discussion of these issues has been added in the updated version of the manuscript (results section in the main text and supplementary material, Section S2. Artifact correction).

4) *The current method is called "fast" but it still requires scanning the sample stepwise. Do you see the potential to do an instantaneous mapping like in standard light microscopy? This may be worth commenting on in the conclusion section.*

Answer. We call SAROM a “fast” method because it can determine the orientation of a crystal in one “shot” due to simultaneous acquisition of nine correctly designed polarized channels. Previously, sequential measurements at different orientation of laser polarization state (or analyzer orientation) were required for orientation determination.

Nevertheless, we think that “semi-quantitative” orientation determination by Raman microscopy in wide-field illumination mode similar to standard light microscopy can be possible.

In order to realize wide field Raman a tunable filter (acoustic-optical tunable filter or interferometric tunable filter) should be used instead of spectrometer. We see several limitations in such wide field polarized Raman microscopy:

- 1) true off-axis registration required for accurate orientation determination is not possible because wide-field imaging mixes pure on-axis and pure off-axis signals;
- 2) multiple polarized wide field maps should be obtained sequentially. Therefore, the technique would not be dramatically faster than the proposed SAROM method;
- 3) wide-field Raman microscopes can collect Raman maps in a selected narrow spectral range. However, accurate orientation determination sometimes requires tensor analysis of

more than one phonon modes. This will require additional measurements by wide-field Raman microscope in a different spectral range which will increase total measurement time.

- 4) spectral resolution. Existing wide-field Raman microscopes provide spectral resolution around $10\text{-}30\text{cm}^{-12-5}$. This can be not enough for independent registration of phonon modes.

We appreciate the reviewer for the proposition of commenting wide-field polarized Raman mapping in the conclusion section. This discussion has been added in the revised version of the manuscript.

References

1. Mizoguchi, K. & Nakashima, S. I. Determination of crystallographic orientations in silicon films by Raman-microprobe polarization measurements. *J. Appl. Phys.* **65**, 2583–2590 (1989).
2. Schlücker, S., Schaeberle, M. D., Huffman, S. W. & Levin, I. W. Raman Microspectroscopy: A Comparison of Point, Line, and Wide-Field Imaging Methodologies. *Anal. Chem.* **75**, 4312–4318 (2003).
3. Papour, A. *et al.* Wide-field Raman imaging for bone detection in tissue. *Biomed. Opt. Express* **6**, 3892 (2015).
4. Ling, J., Weitman, S. D., Miller, M. A., Moore, R. V. & Bovik, A. C. Direct Raman imaging techniques for study of the subcellular distribution of a drug. *Appl. Opt.* **41**, 6006 (2002).
5. Morris, H. R., Hoyt, C. C., Miller, P. & Treado, P. J. Liquid Crystal Tunable Filter Raman Chemical Imaging. *Appl. Spectrosc.* **50**, 805–811 (1996).

REVIEWERS' COMMENTS:

Reviewer #1 (Remarks to the Author):

The authors did a good job in answering the comments and concerns raised by the reviewers in a extensive and satisfactory way. I recommend publication of the manuscript in its present form.

Reviewer #2 (Remarks to the Author):

All points noted in my previous reports have been clarified.

Publication is recommended. Nice work.

Reviewer #3 (Remarks to the Author):

The authors have addressed the points raised and revised their manuscript accordingly. The quality of the paper has been significantly improved and the novelty is pointed out more clearly. Therefore, I recommend publication of the manuscript.